# Sphingolipid Metabolism Is Associated with Cardiac Dyssynchrony in Patients with Acute Myocardial Infarction

**DOI:** 10.3390/biomedicines12081864

**Published:** 2024-08-15

**Authors:** Ching-Hui Huang, Chen-Ling Kuo, Yu-Shan Cheng, Ching-San Huang, Chin-San Liu, Chia-Chu Chang

**Affiliations:** 1Division of Cardiology, Department of Internal Medicine, Changhua Christian Hospital, Changhua 500, Taiwan; 28071@cch.org.tw; 2Department of Mathematics, National Changhua University of Education, Changhua 500, Taiwan; 3Department of Beauty Science and Graduate Institute of Beauty Science Technology, Chienkuo Technology University, Changhua 500, Taiwan; 4Vascular Medicine and Diabetes Research Center, Changhua Christian Hospital, Changhua 500, Taiwan; 97026@cch.org.tw (C.-L.K.); 149951@cch.org.tw (Y.-S.C.); 5Center of Regenerative Medicine and Tissue Repair, Institute of ATP, Changhua Christian Hospital, Changhua 500, Taiwan; 149390@cch.org.tw; 6Department of Neurology, Changhua Christian Hospital, Changhua 500, Taiwan; liu48111@gmail.com; 7Department of Internal Medicine, Kuang Tien General Hospital, Taichung 433, Taiwan; 8Department of Nutrition, Hungkuang University, Taichung 433, Taiwan

**Keywords:** sphingolipids, systolic dyssynchrony index, acute myocardial infarction

## Abstract

Aim: Sphingolipids are a class of complex and bioactive lipids that are involved in the pathological processes of cardiovascular disease. Fabry disease is an X-linked storage disorder that results in the pathological accumulation of glycosphingolipids in body fluids and the heart. Cardiac dyssynchrony is observed in patients with Fabry disease and left ventricular (LV) hypertrophy. However, little information is available on the relationship between plasma sphingolipid metabolites and LV remodelling after acute myocardial infarction (AMI). The purpose of this study was to assess whether the baseline plasma sphingomyelin/acid ceramidase (aCD) ratio predicts LV dyssynchrony at 6M after AMI. Methods: A total of 62 patients with AMI undergoing primary angioplasty were recruited. Plasma aCD and sphingomyelin were measured prior to primary angioplasty. Three-dimensional echocardiographic measurements of the systolic dyssynchrony index (SDI) were performed at baseline and 6 months of follow-up. The patients were divided into three groups according to the level of aCD and sphingomyelin above or below the median. Group 1 denotes lower aCD and lower sphingomyelin; Group 3 denotes higher aCD and higher sphingomyelin. Group 2 represents different categories of patients with aCD and sphingomyelin. Trend analysis showed a significant increase in the SDI from Group 1 to Group 3. Logistic regression analysis showed that the sphingomyelin/aCD ratio was a significant predictor of a worsening SDI at 6 months. Conclusions: AMI patients with high baseline plasma sphingomyelin/aCD ratios had a significantly increased SDI at six months. The sphingomyelin/aCD ratio can be considered as a surrogate marker of plasma ceramide load or inefficient ceramide metabolism. Plasma sphingolipid pathway metabolism may be a new biomarker for therapeutic intervention to prevent adverse remodelling after MI.

## 1. Introduction

Sphingolipids are a complex class of lipids that are the main components of the cell membrane structure. Some sphingolipids are bioactive and are related to the pathological process of cardiovascular disease [1]. Sphingolipids can be classified on the basis of their complexity and the presence of additional functional groups. The main classes include ceramides, sphingomyelins, cerebrosides, gangliosides, and sphingosine-1-phosphate (S1P) [2]. Each class has unique structural and functional characteristics that allow them to play an important role in various cellular processes. Cerebrosides are made up of sphingosine, a fatty acid, and galactose or glucose. Therefore, they resemble sphingomyelins but have a sugar unit in place of the choline phosphate group. In contrast, sphingomyelin contains a ceramide backbone, similar to cerebroside, but with phosphocholine groups instead of sugar groups. Cerebroside is abundant in brain white matter and nerve myelin sheaths and is also present in small amounts in the cell membranes of other tissues [3]. A major sphingolipid metabolite is ceramide, which forms the backbone of all complex sphingolipids. Ceramides are generally upregulated in response to stress, such as during inflammation stimuli and ischemia/reperfusion injury. In response to stress, there are three main pathways for the generation of ceramides. First, the sphingomyelinase pathway uses acid sphingomyelinase to break down sphingomyelin and release ceramide. Second, the de novo pathway creates ceramide from less complex molecules. Finally, ceramide may be produced from sphingosine by sphinganine N-acyltransferase (ceramide synthase) in the salvage pathway [4]. Ceramide can be eliminated from cells by ceramidase activity, phosphorylation to 1-phosphoceramide, or the resynthesis of more complex sphingolipids. More importantly, under conditions of ischemia/reperfusion, ceramide accumulation appears to be primarily associated with acid sphingomyelinase activity resulting from sphingomyelin hydrolysis rather than the induction of de novo ceramide synthesis [5]. The enzyme acid sphingomyelinase catalyses the hydrolysis of sphingomyelin to ceramide [6]. These enzymes are major rapid sources of ceramide production not only during physiological responses to receptor stimulation but also in disease status. Previous work has shown some mechanistic insight into the role of acid sphingomyelinase in mediating cardiovascular disease [7,8,9]. The reduced or absent activity of the acid sphingomyelinase results in an abnormal accumulation of sphingomyelin in various tissues of the body, such as Niemann–Pick disease. Ceramidase is the only known enzyme that hydrolyses pro-apoptotic ceramide, producing sphingosine, which is then phosphorylated by sphingosine kinase to produce the pro-survival molecule sphingosine-1-phosphate. An animal study has shown that the transient alteration of sphingolipid metabolism by the overexpression of acid ceramidase is sufficient and necessary to induce cardioprotection after myocardial infarction [10]. 

Left ventricular dyssynchrony is one of the important predictors of left ventricular remodelling after AMI, and its severity is related to infarct size and left ventricular systolic function [11,12]. In recent years, knowledge and treatments for Fabry disease have evolved greatly. Fabry disease is an X-linked storage disorder caused by a defect in the lysosomal enzyme α-galactosidase A, resulting in the pathological accumulation of glycosphingolipids in body fluids and in multiple tissues and organs throughout the body [13]. Mechanical cardiac dyssynchrony is observed in 76% of patients with Fabry’s disease and left ventricular hypertrophy [14]. However, there is little information on the relationship between plasma sphingolipid metabolites and LV dyssynchrony after AMI.

Sphingomyelin is a fatty substance that is a component of most cell membranes. Sphingomyelin is degraded by sphingomyelinase to produce ceramide. Ceramide is broken down by ceramidase to produce sphingosine, which can be phosphorylated by sphingosine kinase to form sphingosine-1-phosphate (S1P), a potent signalling molecule [2]. Because the catabolic process from sphingomyelin to sphingosine is dynamic and reversible, we hypothesised that the sphingomyelin/acid ceramidase ratio may serve as a surrogate marker for efficient ceramide metabolism or plasma ceramide load. The inefficient processing of ceramides, resulting in elevated plasma sphingolipid levels, would have a more severe cellular lipotoxicity on cellular energy metabolism and increased left ventricular dyssynchrony after AMI. The purpose of this study was to test the hypothesis that the dysregulation of the sphingolipid metabolite after AMI contributes to left ventricular dyssynchrony. 

## 2. Materials and Methods

### 2.1. Subjects and Study Protocol

In this study, we prospectively consecutively recruited 62 de novo acute STEMI patients who underwent a primary percutaneous coronary intervention (PCI) within 12 h after symptom appearance at Changhua Christian Hospital in Taiwan. Patients enrolled between the ages of 18 and 80 provided their informed consent. This study was approved by the Institutional Review Board of Changhua Christian Hospital, with approval number 160804. Before PCI, venous blood was collected. Plasma was collected by centrifugation at 1000× *g* and 4 °C for 10 min, divided into aliquots, and stored at −80 °C until analysis. Before PCI, a venous blood sample was drawn to measure the blood levels of sphingomyelin and acid ceramidase. Baseline lipid and glucose concentrations were measured after an 8 h fast. The fraction of creatinine kinase MB (CK-MB) and troponin I concentration were measured every 4 h until they began to decrease, and the maximum value was recorded. The diagnosis of myocardial infarction (MI) is associated with the release of cTn and is made based on the fourth universal definition of MI [15]. Echocardiography was performed within the first 2 days and 6 months after primary PCI. The echocardiographic study was performed by the same qualified cardiologist, who was unaware of the plasma sphingomyelin and acid ceramidase levels of the patient.

### 2.2. Echocardiographic Assessments

All 2-dimensional echocardiography parameters were measured according to the guidelines of the American Society of Echocardiography [16]. We used a modified Simpson formula to calculate the left ventricular ejection fraction (LVEF) described by the American Society of Echocardiography [16]. We used the iE33 xMATRIX echocardiography system (Philips Medical Systems, Andover, MA, USA) to obtain real-time 3D echocardiography images. The systolic dyssynchrony index (SDI) was defined as the standard deviation of the time to minimum systolic volume of 16 LV segments, expressed as the percentage of the duration of R-R, as described by Kapetanakis et al. [17]. A higher index indicates greater LV dyssynchrony.

### 2.3. Measurement of Sphingomyelin and Acid Ceramidase

Plasma acid ceramidase activity and sphingomyelin activity were measured by an ELISA kit (Mybiosource, San Diego, CA, USA; Cayman Chemical, Ann Arbor, MI, USA), and each experiment was performed twice according to the manufacturer’s instructions.

### 2.4. Statistical Analysis

Data are given as the mean ± SD. We used Student’s t-test to assess differences between AMI subgroups. The Jonckheere–Terpstra trend test was used to analyse the association between baseline acid ceramidase and sphingomyelin concentrations and the SDI at 6 months. The Jonckheere–Terpstra test is similar to the Kruskal–Wallis test, but it is applied to samples with a priori ranking. Logistic regression models were used to assess the independent association between possible predictors and the deterioration of the SDI at 6 months. *p* < 0.05 was considered statistically significant. All statistical analyses were performed with SPSS for Windows (version 15.0, SPSS, Chicago, IL, USA). Since this is a preliminary study, we do not have reference data on acid ceramidase and sphingomyelin concentrations and LV dyssynchrony in patients with STEMI. Therefore, we cannot estimate the sample size or the study power.

## 3. Results

### 3.1. Baseline Concentrations of Acid Ceramidase and Sphingomyelin

There were 62 patients (age, 55.4 ± 10.3 years; male, *n* = 55; female, *n* = 7) in the AMI group and 55 healthy volunteers (age, 55.3 ± 7.4 years old; male, *n* = 47; female, *n* = 8). The baseline plasma concentration of sphingomyelin and acid ceramidase in the AMI group was significantly lower than in the control group (16.8 ± 14.9 vs. 31.17 ± 8.57 ng/mL, *p* < 0.001; 17.0 ± 8.09 vs. 43.48 ± 14.9 ng/mL, *p* < 0.001, Table 1). However, the sphingomyelin/acid ceramidase ratio in the AMI group was significantly higher than that of the control group (1.19 ± 1.15 vs. 0.79 ± 0.32, *p* = 0.010, Table 1).

### 3.2. Comparisons between Patients with Low and High Baseline Plasma Acid Ceramidase Concentrations

The median baseline acid ceramidase concentration was 14.28 ng/mL. Patients with a low baseline acid ceramidase concentration (below the median) had a significantly lower 6M SDI than those with a high baseline acid ceramidase concentration (above the median) (*p* = 0.026, Table 2). However, there were no significant differences in the baseline SDI between the two subgroups in the AMI group. Patients with AMI with low acid ceramidase (defined as below the median) had a higher LVEF at baseline and 6 months of follow-up. Compared to patients with lower acid ceramidase, patients with AMI with higher acid ceramidase had significantly higher hsCRP, oxLDL, and CyPA concentrations and TIMI risk scores and a lower ratio of sphingomyelin/acid ceramidase (Table 2).

**Table 1 biomedicines-12-01864-t001:** Clinical characteristics and biochemical variables of control and AMI groups.

	AMI Group	Control Group	*p*-Value
	*n* = 62	*n* = 54	
Age, year	55.4 ± 10.3	55.3 ± 7.4	0.264
Sex (M/F)	55/7	47/8	0.384
Cholesterol, mg/dL	191.6 ± 46.5	174.9 ± 23.4	0.014 *
HDL-C, mg/dL	41.4 ± 9.8	65.0 ± 14.4	<0.001 **
LDL-C, mg/dL	134.3 ± 38.8	97.2 ± 22.9	<0.001 **
Triglyceride, mg/dL	89.0 ± 117.9	70.6 ± 36.8	0.239
Fasting glucose, mg/dL	156.3 ± 93.2	87.0 ± 7.7	<0.001 **
HbA1C, %	6.5 ± 1.7	5.1 ± 0.3	<0.001 **
BUN, mg/dL	15.9 ± 6.8	12.3± 2.9	0.019 *
Creatinine, mg/dL	1.0 ± 0.3	0.8 ±0.1	<0.001 **
Fibrinogen, mg/dL	452.9 ± 92.5	285.5 ± 51.1	<0.001 **
hsCRP(mg/L)	0.46 ± 0.67	0.05 ± 0.04	0.003 *
oxLDL, mg/dL	65.7± 21.8	51.8± 31.0	0.002 *
MCN (per cell)	118.4 ± 103.4	194.9 ± 119.5	0.003 *
Sphingomyelin (mg/dL)	16.8 ± 14.9	31.17 ± 8.57	<0.001 **
aCD (ng/mL)	17.0 ± 8.09	43.48 ± 14.9	<0.001 **
Sphingomyelin/aCD	1.19 ± 1.15	0.79 ± 0.32	0.010 *
Current smoking, %	39(63%)	2(4%)	<0.001 **

HDL-C, high-density lipoprotein cholesterol; LDL-C, low-density lipoprotein cholesterol; HbA1C, haemoglobin A1c; BUN, blood urine nitrogen; oxLDL, oxidised low-density lipoprotein; MCN, mitochondrial DNA copy number; aCD: acid ceramidase; hsCRP, high-sensitivity C-reactive protein. * *p* < 0.05, Student’s *t*-test. ** *p* < 0.01.

### 3.3. Comparisons between Patients with Low and High Baseline Sphingomyelin Concentrations

The median baseline sphingomyelin concentration was 12.67 ng/mL. Compared to patients with high baseline sphingomyelin concentrations, patients with low baseline sphingomyelin concentrations (below the median) had significantly lower ApoB, cholesterol, oxLDL, and triglyceride concentrations, while oxidised low-density lipoprotein autoantibodies (OLABs) were even higher (Table 3).

### 3.4. Trend Analysis Revealed That SDI Was Positively Proportional to Baseline Acid Ceramidase and Sphingomyelin Concentrations

We divided the patients into three groups according to acid ceramidase and sphingomyelin levels above or below the median. Group 1 indicated lower acid ceramidase (below the median) and lower sphingomyelin (below the median) (*n* = 12). Group 3 indicated higher acid ceramidase and higher sphingomyelin (*n* = 12). Group 2 represented lower acid ceramidase and higher sphingomyelin or higher acid ceramidase and lower sphingomyelin (*n* = 38). Trend analysis showed that the SDI increased from Group 1 to Group 3 in 6 months. (Jonckheere–Terpstra test, *p* = 0.016, Figure 1).

### 3.5. Evaluation of Possible Predictors of Worsening SDI at 6 Months

The worsening of the SDI at 6 months was defined as the SDI at 6 months minus the SDI at baseline ≥0. Through logistic regression analysis, we confirmed that the sphingomyelin/acid ceramidase ratio may be an important predictor (favourable and unfavourable) of the SDI at 6 months of deterioration with odds ratio 2.263 (Table 4).

## 4. Discussion

### 4.1. Sphingomyelin/Ceramide Pathway Metabolism and LV Dyssynchrony

The main finding of our study indicated that the sphingomyelin/ceramide pathway was associated with LV systolic dyssynchrony 6 months after the AMI event. Trend analysis revealed that the SDI was positively proportional to baseline acid ceramidase and sphingomyelin concentrations (Figure 1). Patients with lower baseline acid ceramidase (below the median) and lower baseline sphingomyelin (below the median) had a favourable SDI at 6 months. However, patients with higher baseline acid ceramidase and higher baseline sphingomyelin developed an unfavourable SDI at 6 months. The results are shown in Figure 1. We postulated that sphingomyelin and acid ceramidase levels should be considered simultaneously to reflect the plasma ceramide load or efficient ceramide processing. In the logistic regression model, we considered possible influencing factors for the worsening of the SDI at 6 months based on previous studies. These include the size of the infarct [11,12], expressed as the maximum value of the cardiac enzyme CKMB, the coronary arteries related to the infarct [18], and the levels of CoQ10 associated with oxidative stress and the inflammatory status, expressed as levels of oxLDL and IL-6, respectively [19,20]. Furthermore, the age, sex, and sphingomyelin/acid ceramide ratio were also considered. Through logistic regression analysis, we confirmed that the sphingomyelin/acid ceramidase ratio may be an important predictor of the SDI at 6 months after AMI, with an odds ratio of 2.273 (Table 4), and may be considered a surrogate marker of plasma ceramide load or inefficient ceramide metabolism. To our knowledge, no studies have addressed the relationship between LV dyssynchrony and sphingomyelin/ceramide pathway metabolism in patients with AMI. Due to facility limitations, we did not directly measure plasma ceramide levels. Instead, we measured plasma sphingomyelin and acid ceramidase. The reasons why we chose sphingomyelin and acid ceramidase as targets are explained below in the next paragraph.

### 4.2. The Yin/Yang Concept of Efficient Ceramide Metabolism

First, sphingomyelin is a precursor of ceramide and serves as a substrate for sphingomyelinase, which hydrolyses sphingomyelin to produce ceramide, a bioactive lipid involved in a variety of cellular processes, including apoptosis and inflammation. More importantly, sphingomyelin hydrolysis due to increased sphingomyelinase activity leads to the accumulation of ceramide under ischemia/reperfusion conditions, resulting in increased ceramide levels. Therefore, sphingomyelin may serve as a surrogate marker of plasma ceramide load. Second, acid ceramidase is an enzyme that catalyses the hydrolysis of ceramide to sphingosine, which can then be phosphorylated to form sphingosine-1-phosphate (S1P). S1P has cardioprotective effects, including promoting cell survival, angiogenesis, and anti-inflammatory responses. An animal study demonstrated that transient alterations in sphingolipid metabolism by acid ceramidase overexpression are sufficient and necessary to induce cardioprotection after myocardial infarction [10]. Therefore, acid ceramidase may serve as a surrogate marker for efficient ceramide processing. Our arguments come from two documents [21,22] to support our speculation. Ceramide is undoubtedly a link in critical signalling networks and has a profound impact on a variety of cardiometabolic diseases, cancer, neurodegenerative diseases, infections, inflammatory processes, etc. The core idea is how to reduce ceramide levels and promote its metabolism to synthesise S1P, balancing the pro-survival and pro-apoptotic functions of each sphingolipid (graphical abstract). Therefore, ceramide can be considered as yin; S1P can be considered as yang. Healthy people have a balance of yin and yang, while sick people have an imbalance of yin and yang. As this is a preliminary study, we used cultured epithelial cells from cystic fibrosis patients and improved inflammation and infection in cystic fibrosis lung disease using recombinant acid ceramidase as examples to illustrate our argument [23]. The distribution of ceramide with sphingomyelin and sphingosine contributes to the delicate balance between anti-inflammatory function and antimicrobial efficiency in the host’s response to infection [23,24,25,26]. Lipid modifications induced by different pathogens are important because they have unique exchanges with cell membranes that can trigger local changes in membrane properties and influence downstream signalling. Inefficient ceramide processing results in insufficient sphingosine levels and is associated with the ineffective treatment of infection and inflammation. The disproportionate relationship between sphingomyelin, ceramide, and sphingosine is associated with an increased susceptibility to pulmonary infection and in vivo inflammation associated with cystic fibrosis P. aeruginosa infection [23]. Understanding the specific relationship between the ratios of sphingomyelin, ceramide, and sphingosine is important for the impact on downstream signalling and contributes to the delicate balance between loading and efficient ceramide processing. Since sphingolipid metabolism is dynamic, the physiological functions of sphingolipids are strictly dependent on their concentration [27]. We postulated that in left ventricular remodelling, the simultaneous evaluation of plasma metabolite relationships during sphingolipid metabolism would be more important than the examination of individual bioactive sphingolipids. We hypothesised that the ratio of sphingomyelin to acid ceramidase could serve as a surrogate marker of plasma ceramide load or efficient ceramide metabolism. Inefficient ceramide processing, which results in insufficient levels of sphingosine, may be associated with adverse cardiac remodelling parameters.

### 4.3. A Possible Reason for Lipid Metabolism and Heart Dyssynchrony

Left ventricular dyssynchrony is one of the important predictors of left ventricular remodelling after AMI, and its severity is related to the size of the infarct and left ventricular systolic function [11,12]. Previous studies have shown that cardiomyocytes increase intracellular lipids due to metabolic remodelling after MI [28,29]. A lipidomic analysis of the human heart showed that the main lipids present in infarcted human heart tissue are sphingolipids and certain types of cholesterol [30]. The authors found that there is a strong positive correlation between human heart lipids and adverse cardiac remodelling after MI. By analysing the plasma metabolism of MI patients, sphingolipid metabolism is the most altered pathway in patients with STEMI [31] and may represent a valuable prognostic factor and potential therapeutic target [32]. Furthermore, a recent porcine study showed that the pacing of the right ventricle inhibits lipid metabolic pathways and leads to the accumulation of intracellular lipids. Lipotoxicity was induced by the pacing of the right ventricle associated with persistent left ventricular dyssynchrony [33]. Furthermore, in recent years, the knowledge and treatments for Fabry’s disease have evolved greatly. Fabry disease is a rare X-linked genetic disorder caused by mutations in the GLA gene encoding α-galactosidase A (α-Gal A). Alpha-Gal A deficiency results in the accumulation of triacylceramide (Gb3 or GL-3) and related glycosphingolipids in body fluids and various tissues, including the heart [13]. This accumulation causes cardiomyocyte hypertrophy, fibrosis, and endothelial dysfunction, contributing to electrical and mechanical dyssynchrony. Mechanical cardiac dyssynchrony is observed in 76% of patients with Fabry’s disease and left ventricular hypertrophy [14]. The resulting disruption of normal electrical conduction and impaired synchronous contraction of the heart muscle lead to the clinical manifestations of cardiac dyssynchrony in Fabry’s disease. These findings emphasise the importance of lipid metabolism and heart dyssynchrony. The above studies are consistent with our study; that is, sphingolipid metabolism is related to LV systolic dyssynchrony and can serve as a biomarker for the probability of adverse outcomes after AMI. Monitoring these biomarkers can help to stratify risk and tailor personalised therapeutic strategies.

### 4.4. Plasma Levels of Acid Ceramidase and Sphingomyelin in AMI Patients

During the development of ischemia/reperfusion damage, the initial response of reactive oxygen species and Tumour Necrosis Factor-α will activate the two main ceramide production pathways (sphingomyelin hydrolysis and de novo ceramide synthesis). Depending on the stage of ischemia/reperfusion, increased ceramide has a wide range of effects, including pro-apoptotic and anti-apoptotic effects [4]. Therefore, strategies to reduce ceramide levels, for example, by modulating the activity of ceramidase and/or sphingomyelinase, may represent a new and promising therapeutic approach to prevent or treat ischemia/reperfusion injury in different clinical settings [34]. In patients with AMI, a decrease in acid ceramidase levels leads to an increase in ceramide. Furthermore, a transient increase in acid ceramidase is sufficient to induce cardioprotection after myocardial infarction [10]. Several studies have evaluated the association between circulating plasma levels of sphingomyelin and the incidence of cardiovascular disease, with contradictory results [35]. In our study, compared to the control group, plasma acid ceramidase and sphingomyelin were significantly lower in the AMI group. However, sphingomyelin/acid ceramidase was significantly higher in the AMI group (1.19 ± 1.15 vs. 0.79 ± 0.32, *p* = 0.010). These findings indicate that the activity of acid ceramidase is downregulated, sphingomyelinase is upregulated, and sphingomyelin hydrolysis is enhanced to produce ceramide. The ratio of sphingomyelin to acid ceramidase may be considered a surrogate marker of plasma ceramide load or inefficient ceramide metabolism, but more clinical studies are needed to confirm our findings.

### 4.5. Research Strengths and Future Research Directions

In recent years, through innovative improvements in mass spectrometry (MS) and chromatographic techniques, lipidomics can better identify the composition of lipid molecular species, including ceramides, in biological samples [36]. In clinical laboratories, the use of LC-MS/MS is growing, with high specificity and sensitivity. However, these techniques are not easily implemented in all clinical laboratories, as they remain expensive, require specialised and sophisticated instrumentation, and require careful maintenance, skilled operators, and technical expertise [36]. Given its ubiquitous nature and rapid deregulation during the onset and progression, the practical use of a single specific ceramide is highly impractical, especially given the intra- and inter-subject heterogeneity and the specific pathophysiological effects of each retention of ceramide [37]. Therefore, simultaneously considering ceramide and its precursors and downstream metabolites, such as the ratio of sphingomyelin/acid ceramide in our study, is a macroscopic perspective and a feasible approach in clinical practice. However, more studies must be conducted to make this information more useful in clinical settings. Further research should focus on conducting interventional trials to evaluate the impact of sphingolipid-targeted therapy on patient recovery and survival, using techniques such as Mediterranean or Nordic dietary interventions, exercise interventions, or drug interventions targeting sphingolipid metabolism.

### 4.6. Study Limitations

This study has several limitations. First, due to facility limitations, we did not directly measure plasma ceramide levels. Instead, we measured plasma sphingomyelin and acid ceramidase and calculated the sphingomyelin/acid ceramidase ratio as a surrogate marker of plasma ceramide load or inefficient ceramide processing. More clinical studies are needed to confirm our findings. Secondly, the number of female patients was relatively small, but we enrolled consecutive cases within a given time period, which is consistent with real-world male-to-female AMI ratios in our institute. Therefore, a larger sample size with more data and a balanced ratio of men and women is needed in future studies to validate our finding. Third, this study has a short follow-up time for the sample, and a longer study time is needed to validate the conclusions of our study. Fourth, although possible influencing factors have been considered as much as possible, there are still other factors that have not been considered for the deterioration of the SDI at 6 months; more research should be conducted to reconfirm our findings.

## 5. Conclusions

Patients with AMI with a high baseline plasma sphingomyelin/acid ceramidase ratio had a significant increase in the systolic dyssynchrony index at 6 months. The dysregulation of sphingolipid pathways can contribute to cardiac dyssynchrony. The sphingomyelin/acid ceramidase ratio may be considered as a surrogate marker of plasma ceramide load or inefficient ceramide metabolism. Plasma sphingolipid pathway metabolism may be a new biomarker for therapeutic intervention to prevent adverse remodelling after MI.

## 6. Clinical Perspectives

Patients with AMI with a higher baseline plasma sphingomyelin/acid ceramidase ratio had a significantly higher SDI at six months.We demonstrated that the ratio of sphingomyelin/acid ceramidase can be considered as a surrogate marker of plasma ceramide load or inefficient ceramide metabolism.Plasma sphingolipid pathway metabolism may be a new biomarker for therapeutic intervention to prevent adverse remodelling after AMI.

## Figures and Tables

**Figure 1 biomedicines-12-01864-f001:**
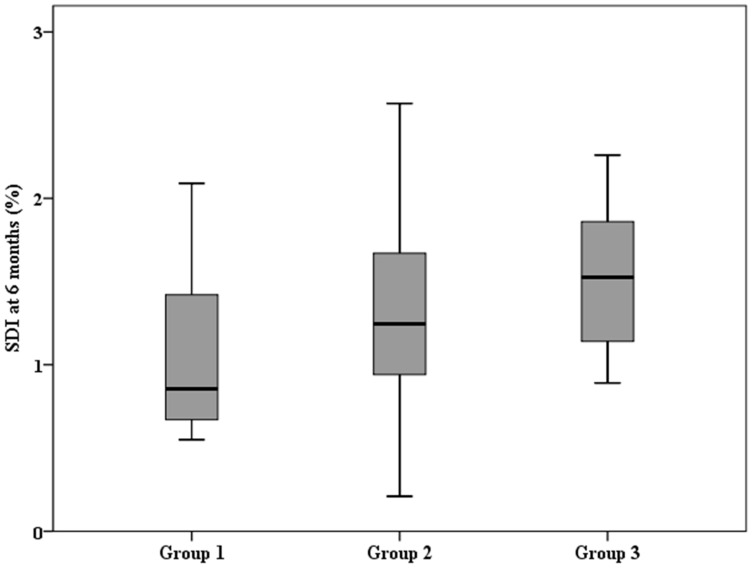
Trend analysis revealed that the six-month SDI was positively proportional to baseline acid ceramidase and sphingomyelin concentrations. Group 1 indicated lower acid ceramidase (below the median) and lower sphingomyelin (below the median) (*n* = 12). Group 3 indicated higher acid ceramidase and higher sphingomyelin (*n* = 12). Group 2 represented lower acid ceramidase and higher sphingomyelin or higher acid ceramidase and lower sphingomyelin (*n* = 38). (Jonckheere–Terpstra test, *p* = 0.016).

**Table 2 biomedicines-12-01864-t002:** Comparisons between patients with low and high baseline plasma acid ceramidase concentrations.

	Low Baseline aCD	High Baseline aCD	
	(≤14.28 ng/mL; *n* = 31)	(>14.28 ng/mL; *n* = 31)	*p*-Value
Age (years)	55.25 ± 11.93	60.58 ± 11.59	0.073
Sex (Male/female)	29/2	26/5	0.082
CPK, μ/L	1995.84 ± 1742.19	2684.59 ± 2269.83	0.178
CKMB, ng/mL	204.71 ± 174.80	260.22 ± 206.84	0.248
Troponin I, ng/mL	4.04 ± 8.62	6.70 ± 20.21	0.545
Creatinine, mg/dL	1.04 ± 0.38	0.96 ± 0.22	0.357
Cholesterol, mg/dL	189.61 ± 41.88	193.48 ± 51.16	0.742
HDL-C, mg/dL	40.81 ± 9.53	42.03 ± 10.18	0.624
LDL-C, mg/dL	132.76 ± 36.66	135.78 ± 41.20	0.756
Triglyceride, mg/dL	97.94 ± 152.83	80.36 ± 70.67	0.552
Fasting glucose, mg/dL	144.23 ± 86.81	168.29 ± 99.11	0.313
HbA1C, %	6.20 ± 1.43	6.79 ± 1.94	0.164
BMI, kg/m^2^	24.99 ± 3.07	26.24 ± 3.95	0.172
Risk factors			
Hypertension	26(84%)	20(65%)	0.039 *
Diabetes mellitus	7(23%)	12(39%)	0.062
Dyslipidaemia	30(96%)	28(90%)	0.310
Smoking	21(68%)	18(58%)	0.880
Infarct-related artery			0.774
LAD	17(53%)	18(60%)	
LCX	5(16%)	3(10%)	
RCA	10(31%)	9(30%)	
SDI baseline (%)	1.11 ± 0.65	1.58 ± 1.21	0.079
SDI 6M (%)	1.24 ± 0.59	2.19 ± 1.97	0.026 *
LVEF baseline (%)	63.6 ± 8.7	57.5 ± 12.1	0.038 *
LVEF 6M (%)	65.8 ± 10.5	59.1 ± 11.8	0.031 *
hsCRP (mg/L)	0.25 ± 0.29	0.64 ± 0.86	0.022 *
oxLDL (mg/dL)	57.64 ± 19.62	72.18 ± 22.79	0.046 *
CyPA (ng/mL)	52.97 ± 20.37	67.06 ± 26.90	0.021 *
TIMI risk score	2.41 ± 0.87	3.21 ± 1.15	0.007 **
Sphingomyelin (mg/dL)	17.23 ± 10.38	16.40 ± 18.47	0.833
Sphingomyelin/aCD	1.68 ± 1.32	0.70 ± 0.69	0.001 **

LAD: left anterior descending coronary artery; RCA: right coronary artery; LCX: left circumflex coronary artery; SDI: systolic dyssynchrony index; LVEF: left ventricular ejection fraction; hsCRP: high-sensitivity C-reactive protein; oxLDL: oxidised low-density lipoprotein; CyPA: cyclophilin A; TIMI: thrombolysis in myocardial infarction; aCD: acid ceramidase; * *p* < 0.05, Student’s *t*-test. ** *p* < 0.01.

**Table 3 biomedicines-12-01864-t003:** Comparisons between patients with low and high baseline sphingomyelin concentrations.

	Low Baseline Sphingomyelin	High Baseline Sphingomyelin	
	(≤12.67 mg/dL; *n* = 31)	(>12.67 mg/dL; *n* = 31)	*p*-Value
Age (years)	61.10 ± 12.42	55.17 ± 11.51	0.060
Sex (Male/female)	27/4	28/3	0.723
CPK, μ/L	2307.03 ± 2021.92	2200.52 ± 2059.64	0.842
CKMB, ng/mL	223.47 ± 177.35	225.31 ± 204.18	0.970
Troponin I, ng/mL	5.39 ± 11.04	6.14 ± 20.43	0.876
Creatinine, mg/dL	1.00 ± 0.25	1.02 ± 0.38	0.826
Cholesterol, mg/dL	175.59 ± 38.06	203.97 ± 51.51	0.020 *
HDL-C, mg/dL	41.53 ± 10.34	41.79 ± 8.49	0.920
LDL-C, mg/dL	122.79 ± 31.05	142.09 ± 44.40	0.056
Triglyceride, mg/dL	56.37 ± 33.23	124.17 ± 164.38	0.034 *
Fasting glucose, mg/dL	150.46 ± 94.49	153.90 ± 85.14	0.886
HbA1C, %	6.44 ± 1.75	6.39 ± 1.48	0.912
BMI, kg/m^2^	24.81 ± 3.04	26.00 ± 4.03	0.217
Risk factors			
Hypertension	20(65%)	26(84%)	0.039 *
Diabetes mellitus	9(29%)	10(32%)	0.787
Dyslipidaemia	28(90%)	30(97%)	0.310
Smoking	19(61%)	20(65%)	0.603
Infarct-related artery			0.588
LAD	17(54%)	18(58%)	
LCX	4(13%)	4(13%)	
RCA	10(33%)	9(29%)	
LVEF baseline (%)	60.17 ± 11.62	63.12 ± 9.22	0.318
LVEF 6M (%)	62.35 ± 12.21	64.61 ± 10.58	0.482
SDI baseline (%)	1.55 ± 1.30	1.14 ± 0.62	0.149
SDI 6M (%)	1.96 ± 2.10	1.36 ± 0.52	0.183
Apo B	89.1 ± 17.33	103.73 ± 24.25	0.010 *
oxLDL (mg/dL)	57.64 ± 19.62	72.18 ± 22.79	0.017 *
OLAB (U/L)	392.62 ± 403.44	217.47 ± 177.88	0.048 *
aCD (ng/mL)	16.99 ± 6.29	16.79 ± 10.04	0.928
Sphingomyelin/aCD	0.50 ± 0.35	1.88 ± 1.26	<0.001 **

LAD: left anterior descending coronary artery; RCA: right coronary artery; LCX: left circumflex coronary artery; SDI: systolic dyssynchrony index. LVEF: left ventricular ejection fraction; oxLDL: oxidised low-density lipoprotein; OLAB: oxidised low-density lipoprotein antibody; aCD: acid ceramidase; * *p* < 0.05, Student’s *t*-test. ** *p* < 0.01.

**Table 4 biomedicines-12-01864-t004:** Logistic regression analysis of predictors of worsening SDI at 6 months.

	B	SE	*p*-Value	Odds Ratio	95%CI	
Age	0.016	0.032	0.622	1.016	0.954	1.082
Gender	−0.619	1.094	0.572	0.59	0.063	4.594
oxLDL	−0.009	0.020	0.678	0.992	0.953	1.032
CoQ10	−3.501	3.973	0.378	0.030	0.000	72.637
CKMB mass	0.003	0.002	0.194	1.003	0.999	1.007
Sphingomyelin/aCD	0.817	0.393	0.038 *	2.263	1.047	4.890
IRA	−0.291	0.397	0.463	0.747	0.343	1.626
IL-6	0.070	0.042	0.095	1.073	0.988	1.165
Constant	−0.964	3.266	0.768	0.381		

* *p* < 0.05. We tested sphingomyelin/aCD as a covariate; aCD: acid ceramidase; IRA: infarct-related coronary artery; oxLDL: oxidised low-density lipoprotein.

## Data Availability

The original contributions presented in this study are included in this article; further inquiries can be directed to the corresponding author.

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
