# Peer review of "Sphingolipid Metabolism Is Associated with Cardiac Dyssynchrony in Patients with Acute Myocardial Infarction"

_biomedicines, 2024, doi:10.3390/biomedicines12081864_

Round 1

Reviewer 1 Report

Comments and Suggestions for Authors

What is the  role of Acid sphingomyelinase (ASMase) on  role of sphingomyelin? How does this factor influence the parameters under study in the patients concerned? 

What is the main difference between sphingomyelin and cerebroside concerned with the present study, how does the role of sugar patients contribute with the levels of cholesterol and blood pressure? Moreover, the cases under study were affected by COVID-19,  if so, what could be accounted for

The authors have correlated the level of smoking with the heart disease and level of various disorders, in general will the level of alcohol intake combined with smoking aggregate the disorder?  Had the authors  considered the factor in their study

Author Response

Thank you very much for your reviewing process of our manuscript “Sphingolipids metabolism is associated with cardiac dyssynchrony in patients with acute myocardial infarction” (biomedicines-3094412). The comments raised by the reviewers and Academic Editor were helpful and have been integrated into this revised submission. We appreciate the editor’s and reviewers’ comments to improve the readability of the manuscript; each of their points has been addressed. Revised portions are highlighted in red in the revised manuscript.

Reviewer 2 Report

Comments and Suggestions for Authors

This study is designed to evaluate whether the baseline plasma sphingomyelin/acid ceramidase (aCD) ratio predicts LV asynchrony at 6 months after AMI. These results suggest that the sphingomyelin/aCD ratio can be used as a surrogate marker for plasma ceramide burden or inefficient ceramide metabolism, and plasma sphingolipid pathway metabolism may be a new biomarker for therapeutic interventions to prevent adverse remodeling after myocardial infarction. However, the article still has the following problems:

1.     The sample size of the study is small, and the sex ratio of the selected population is imbalanced (the proportion of female patients is small), so more sample data and a balanced ratio of men and women are needed to explain.

2.     This study has a short follow-up time for the sample, and a longer study is needed to validate the conclusions of the article.

3.     The article suggests that "the ratio of sphingomyelin to acid ceramide can be considered as a surrogate indicator of plasma ceramide load or inefficient ceramide metabolism", and more evidence should be given to support this view.

4.     More research should be done to assess possible factors for the deterioration of SDI at six months to exclude the influence of other factors on the conclusions of the article.

5.    There are too many keywords in the article.

Author Response

(The authors gave the same response as above.)

Round 2

Reviewer 2 Report

Comments and Suggestions for Authors

The author has provided answers to my questions. The limitations section of the article describes the shortcomings that I have pointed out. It's interesting of “The Yin/Yang concept of efficient ceramide metabolism”. I am satisfied with the author's response and the revised manuscript.